



**Estimation of soil water holding capacity with Random Forests for**
**drought monitoring**

Yves Tramblay [1*]
Pere Quintana Seguí [2]
1 HydroSciences Montpellier (University Montpellier, CNRS, IRD), France
2 Observatori de l'Ebre (OE), Ramon Llull University – CSIC, 43520 Roquetes, Spain
*Corresponding author : yves.tramblay@ird.fr, 300 avenue du Pr. Emile Jeanbreau,
34090, Montpellier, France. +33 4 67 14 33 59
**Abstract**
Soil moisture is a key variable for drought monitoring but soil moisture measurements
networks are very scarce. Land-surface models can provide a valuable alternative to
simulate soil moisture dynamics, but only a few countries have such modelling schemes
implemented for monitoring soil moisture at high spatial resolution. In this study, a soil
moisture accounting model (SMA) was regionalized over the Iberian Peninsula, taking as
a reference the soil moisture simulated by a high-resolution land surface model. To
estimate soil water holding capacity, the parameter required to run the SMA model, two
approaches were compared: the direct estimation from European soil maps using
pedotransfer functions, or an indirect estimation by a Machine Learning approach,
Random Forests, using as predictors altitude, temperature, precipitation,
evapotranspiration and land use. Results showed that the Random Forest model
estimates are more robust, especially for estimating low soil moisture levels.
Consequently, the proposed approach can provide an efficient way to simulate daily soil
moisture and therefore monitor soil moisture droughts, in contexts where high-resolution
soil maps are not available, as it relies on a set of covariates that can be reliably estimated
from global databases.
**Keywords:** soil moisture, droughts, random forests




## 1. Introduction

Soil moisture droughts have strong impacts on vegetation and agricultural production (Raymond et al., 2019; Tramblay et al., 2020; Vicente-Serrano et al., 2014; Pena-Gallardo et al., 2019). There is a growing interest for simple indicators to monitor drought events at short timescales that could be related to impacts (Li et al., 2020; Noguera et al., 2021). In particular, soil moisture indicators could be more relevant than climatic ones to monitor potential impacts of droughts on agriculture and natural vegetation (Piedallu et al., 2013). Since actual soil moisture measurements remain very scarce, soil moisture simulated from land-surface models are an interesting proxy to develop simplified methodologies that could be applied on data-sparse regions. Land-surface models (LSM) are valuable tools for a fine scale monitoring of drought events; however, their implementation requires accurate forcing data and computational resources (Almendra-Martín et al., 2021; Quintana-Seguí et al., 2019; Barella-Ortiz and Quintana-Seguí, 2019). Global implementation also exists but with a coarser resolution and driven by reanalysis data (Rodell et al., 2004; Muñoz Sabater, 2020) that may not be adequate for local-scale applications. Only very few countries have land-surface schemes implemented at the national level to monitor droughts (Habets et al., 2008).

Remote Sensing is another option which allows monitoring soil moisture (Dorigo et al., 2017; Brocca et al., 2019). Microwave sensors allow monitoring of surface soil moisture (first 5 cm for L-band based products, skin for C-band based products), without the interference of clouds. However, surface soil moisture is not enough for most applications, which require root zone soil moisture, which is the water resource in the soil available to plants. Furthermore, passive L-band products, such as SMOS (Martínez-Fernández et al., 2016) or SMAP (Mishra et al., 2017), have a low resolution and active C-band products, such as Sentinel 1 (Bauer-Marschallinger et al., 2019), which have higher resolution, suffer from higher noise and are more sensitive to vegetation. Thus, even though remote sensing is very useful, it still has problems to be surmounted. The resolution of passive L-band products can be increased using optical data (NDVI, LST), by means of downscaling algorithms (Merlin et al., 2013; Fang et al., 2021), but then the resulting product is sensitive to cloud cover. Also, some progress has been made in deriving root zone soil moisture from surface soil moisture estimations using an exponential filter (Stefan et al., 2021) calibrated using the SURFEX LSM (Masson et al., 2013), but these products are in early stages and are not operational yet.

Simplified methodologies to estimate and monitor the status of soil moisture, are needed in contexts where LSM data is not available and where remote sensing products fall short, such as areas and time periods with dense vegetation, or high soil roughness which may affect their accuracy (Escorihuela and Quintana-Seguí, 2016). Different modelling





approaches have been proposed, either with conceptual soil moisture accounting models
or computational variants of the antecedent precipitation index (Willgoose and Perera,
2001; Javelle et al., 2010; Brocca et al., 2014; Zhao et al., 2019; Li et al., 2020). The
general availability of spatial estimates of soil moisture content would help introduce soil
moisture in drought monitoring systems, improving their scope and usefulness.
Furthermore, this would also facilitate the creation of long-term reanalysis, based on
meteorological forcing data, and future climate change studies, without the need of
running LSM models. However, to apply this type of models at regional or national scale,
there is a need to estimate their parameters over the area of interest. For that purpose,
regionalization methods have been employed in hydrology for decades to estimate the
parameters of hydrological models in ungauged basins (Blöschl and Sivapalan, 1995; He
et al., 2011; Hrachowitz et al., 2013). Several methods exist, based either on catchment
similarity or the direct estimation of model parameters using regression techniques with
physiographic attributes. For soil moisture modelling, up to now only very few studies
have considered these approaches to apply soil moisture accounting models at ungauged
locations (Grillakis et al., 2021) or estimate root zone soil moisture using machine learning
methods (Carranza et al., 2021).
The goal of the present study is to regionalize (ie. to estimate from surrogate data without
calibration) the soil water holding capacity that is the sole parameter required in a simple
soil moisture model to monitor soil moisture droughts. Two different approaches are
compared: the direct estimation of the soil water holding capacity with soil maps or an
estimation with machine learning techniques, namely Random Forests.
**2. Study area and Data**
The study area of this work is the Iberian Peninsula, which is located between the
Mediterranean Sea and the Atlantic Ocean and thus is influenced by both synoptic scale
systems, that often come from the Atlantic side, and mesoscale heavy precipitation
events, that often come from the Mediterranean side. The Iberian Peninsula presents a
marked relief, with a large and high central plateau and different mountain ranges, which
heavily influence the spatial patterns of precipitation, enhancing it windward and
decreasing it leeward, generating areas of high precipitation on the west, north-west and
north, and very dry areas on the central plains and, specially, on the South-east, as a
consequence the Iberian Peninsula has a heterogeneous distribution of average annual
rainfall, with values ranging from 2000 mm/y to less than 100 mm/y. All this has a strong
influence on the spatial and temporal variability of soil moisture and soil moisture regimes,
having wet regimes on the west and north, where the soil is hardly stressed and, and
semi-arid areas elsewhere, with a wet (energy limited) and a dry (water limited) season,
with a dry down that might be interrupted by convective events. All this makes the
modelling of soil moisture in Iberian a rather challenging task.
Daily precipitation, temperature and evapotranspiration were retrieved from the SAFRAN-
Spain database (Quintana-Seguí et al., 2017). SAFRAN (Durand et al., 1993) is a
meteorological reanalysis that produces gridded datasets by combining the outputs of a
meteorological model and all available observations using an optimal interpolation
algorithm. It has been implemented over France (Quintana-Seguí et al., 2008) and
recently over the Iberian Peninsula (Quintana-Seguí et al., 2017) with a 5kmx5km spatial
resolution. The SAFRAN dataset used in this study not only includes observations from
the Spanish part of the Iberian Peninsula, it has also ingested data from Portugal. The
SURFEX LSM (Masson et al., 2013) has been run using SAFRAN-Spain as the
meteorological forcing dataset and on the same grid, as it was done in Quintana-Seguí
et al., (2020). SURFEX uses the ECOCLIMAP2 (Faroux et al., 2013) physiographic
database and it uses the ISBA (Interaction Sol-Biosphère-Atmosphère) scheme (Noilhan
and Mahfouf, 1996) for natural surfaces. ISBA has different options; we have used ISBA-
DIF, the multi-layer diffusion version (Boone 2000; Habets et al. 2003). From this
simulation, we have extracted the soil moisture of the first 60 cm of the soil, by performing
the weighted average of the soil layers that fall within this range. This simulated soil
moisture over the Iberian Peninsula is considered herein as the observed reference, in
the absence of dense monitoring networks of soil moisture (Martínez-Fernández et al.,
2015). From the ECOCLIMAP2 database, elevation and land cover data have also been
retrieved and aggregated in the following nine categories: water, bare, ice/snow, urban,
forest, grass, dry crops, irrigated crops, wetlands.
We use the European Soil database (ESDB) produced by the European Soil Data Centre
(Panagos et al., 2012). The ESDB contains information on soil characteristics, including
soil depth and texture for topsoil (0-30cm) and subsoil (30-70cm) layers at a grid
resolution of 1 km. The total available water content (TAWC) is a volumetric parameter
describing the water content between field capacity and permanent wilting point, as a
function of available water content, presence of coarse fragments and depth (Reynolds
et al., 2000). In ESDB, water content at field capacity and permanent wilting point were
determined following the equation from (van Genuchten, 1980) to estimate the soil water
retention curve (Hiederer, 2013). The parameters of the equation are provided by a
pedotransfer function (Wösten et al., 1999) for volumetric soil water content computed
from the soil water retention curve. The pedotransfer function uses soil texture, organic
carbon content and bulk density to determine the parameters of the soil water retention
curve (Hiederer, 2013). In the present work, the TAWC of subsoil and topsoil layers have
been added and averaged at the scale of 5km x 5km, matching the spatial resolution of


the SAFRAN grid. Then, these estimates have been used to set the A parameter of the
SMA model.

### 3. Methods

#### 3.1 Soil moisture accounting model

We use a soil moisture accounting model (SMA) driven by precipitation and PET, with
one single parameter A, representing the soil water holding capacity. The soil moisture
model considered here has been previously applied in several studies for applications
related to soil moisture monitoring (Anctil et al., 2004; Javelle et al., 2010; Tramblay et
al., 2012, 2014), it consists in the SMA part of the GR4J model  (Perrin et al., 2003). The
output of the model is daily normalized soil moisture, allowing to detect the days close to
saturation (1) or to complete soil moisture depletion (0).
The SMA model is calibrated using soil moisture simulated with SURFEX covering the
full Iberian Peninsula domain. The Nelder-Mead simplex algorithm is used for the
calibration with the Nash efficiency criterion. The outputs of SURFEX soil moisture are
first normalized with the maximum and minimum values prior to the calibration to compute
the SWI consistent with the SMA model output. To regionally estimate the values of A,
two different methods are compared: the direct estimation of A with TAWC from ESDB
soil maps or its indirect estimation with machine learning methods, namely Random
Forests using 5kmx5km grid physiographic properties.

#### 3.2 Random forests for regionalization of soil water holding capacity

Random Forests (Breiman, 2001) belong to the class of Machine Learning techniques.
RF are based on a bootstrap aggregation (Breiman, 1996) of Classification and
Regression Trees (Breiman et al., 2017). It generates a bootstrap sample from the original
data and trains a tree model using this sample. The procedure is repeated many times
and the bagging's prediction is the average of the predictions. Among the many
advantages of RF, they are fast, non-parametric, robust to noise in the predictor variables,
able to capture nonlinear dependencies between predictors and dependent variables and
they can simultaneously incorporate continuous and categorical variables (Tyralis et al.,
2019). The drawbacks are they are complex to interpret and they cannot extrapolate
outside the training range. Given their advantages, this algorithm is particularly suited for
the estimation of spatial variables such as soil properties (Booker and Woods, 2014;
Hengl et al., 2018; Gagkas and Lilly, 2019; Stein et al., 2021). In the present work, a RF
model is generated to estimate the values of the A parameter of the SMA model,




representing soil water holding capacity, with the properties of the 5x5km grid cells using
Random Forests.

To estimate the reliability of the method, the 5km x 5km grid cells covering the Iberian
Peninsula have been split randomly into a training sample containing 70% of the cells
and a testing sample with the 30% remaining cells. The random selection of the training
and testing sets have been performed using a Latin Hypercube Sampling (McKay et al.,
1979) to ensure a homogeneous sampling over the Iberian Peninsula. Given that the RF
trees cannot be interpreted directly, as for example the weights in a linear regression, we
additionally implemented an out-of-bag predictor importance estimation by permutation
(Loh and Shih, 1997), to measure how influential the predictor variables in the model are
at predicting the response. The influence of a predictor increases with the value of this
measure. If a predictor is influential in prediction, then permuting its values should affect
the model error. If a predictor is not influential, then permuting its values should have little
to no effect on the model error.

**3.3 Validation on the ability to detect dry soil moisture conditions**

To compare the efficiency of the two methods compared to estimate the A parameter of
the SMA model, the SMA model was run using the two methods and all daily values of
soil moisture below the 10th percentile were extracted, corresponding to dry soil
conditions. Only the grid cells in the testing sample were considered for this validation.
We computed different verification scores to assess the relative efficiency of the two
methods to reproduce daily soil moisture below the 10th percentile using the ISBA
simulated soil moisture as a benchmark; the Probability of Detection (POD), the False
Alarm Ratio (FAR) and the Heidke Skill Score (HSS) summarizing the global efficiency to
detect dry periods (Jolliffe and Stephenson, 2011). These scores are based on the
contingency table between forecasts (or simulated values in the case of the present
study) and observations (Table 1).

POD is the probability of detection (equation 1), FAR is the number of false alarms per
the total number of warnings or alarms (equation 2) and HSS is a skill score ranging from
-∞ to 1 (equation 3), for categorical forecasts where the proportion of correct measure is
scaled with the reference value from correct forecasts due to chance.

$POD = a / (a + c)$                                          eq.1

$FAR = b / (a + b)$                                          eq.2

$HSS = 2 (ad - bc) / (a + b)(b + d) + (a + c)(c + d)$         eq.3



## 4. Results

### 4.1 Calibration of the SMA model

The calibration results of the SMA model against SURFEX soil moisture provide very good model performance, with a mean Nash coefficient equal to 0.94, indicating its ability to reproduce the soil moisture dynamics as simulated by SURFEX. Nash values below 0.5 are found for 1.21 % of grid cells (n= 273), only for areas located in the mountainous range affected by snow processes, above 1500 m.a.s.l. (Figure 1). This outcome is expected, since the SMA model does not include a snow-module it cannot reproduce snow dynamics in these areas. However, high-elevation areas with seasonal snow cover are not the area's most at risk of soil moisture droughts for agricultural activities in Spain. The calibrated values of the A parameter of the SMA model ranges from 60 to 250 mm, depending on the location (Figure 3). There is no significant correlation between A and mean annual precipitation or the aridity index (P/PET). This highlights the interplays between soil properties and climate to explain the spatial variability on soil water holding capacity.

### 4.2 Regional estimation of the A parameter

The values of the calibrated A parameter are related to the properties of the 5x5km grid cells using Random Forests. First, an out-of-bag predictor importance estimation by permutation is applied to compute the overall performance of RF and estimate the relative influence of each predictor. When using the A estimates in cross-validation to run the SMA model, the loss of performance is very small, the decrease in Nash values in validation is on average equal -0.0019 (with a maximum decrease of -0.04). This is due to the small sensitivity of the SMA model to the value of A, given that the error in the estimation of A is in the range of 10 mm (RMSE = 13.18 mm). This type of validation mimics the case when the estimation at one single location is required, yet since all the remaining points are used for the estimation, it makes the approach in that case very robust. The relative importance for each predictor is plotted on Figure 3, indicating that precipitation and evapotranspiration are two most important predictors, followed by altitude. On the contrary, the land cover attributes for each grid cell are the least important predictors, and removing them from the RF model does not significantly change the results.

To estimate the robustness of the method, we applied a split-sample validation into a testing and a training sample. 70% of the grid cells (15636 data points) were selected for




training the RF model, and the remaining 30% (6701 data points) for testing. The results
are presented for the testing set (Figure 4). The performance in terms of Nash for the
SMA model with A estimated by Random Forests or soil map is very similar, with mean
Nash equal to 0.86 (median = 0.89) with RF and 0.81 (median = 0.85) with soil maps. The
Nash values in validation (testing set) are low, or even negative, only for mountainous
ranges, as expected. Overall, the spatial patterns of the Nash coefficients obtained with
RF or ESDB are very similar too. There are no significant relationships between model
efficiency and the aridity index or the presence of irrigated areas, as identified in the
ECOCLIMAP2 land cover database.
**4.3 Estimation of dry soil conditions**
A further validation is made for daily soil moisture below the 10th percentile corresponding
to dry soil conditions. We computed the Probability of Detection (POD), the False Alarm
Ratio (FAR) and the Heidke Skill Score (HSS) summarizing the global efficiency to detect
dry periods. For both approaches to estimate A, the mean POD is very high, close to
97%, while the FAR is close to 3%. But these average results hide some discrepancy in
the different regions (Figure 5 and 6): the efficiency is the highest for the North-Western
region, the wettest areas of Spain, while in the South and Central parts of Spain the
performance is lower on average. For the wettest parts of the Iberian Peninsula, the POD
remains higher than 94% and the FAR lower than 6% and it is the region where the main
improvements with RF are observed. On average, the RF estimation method outperforms
the approach based on ESDB (Figure 7), with more stable results in terms of HSS since
all values obtained with RF are above 0.4 while with ESDB for the grid cells the HSS
scores drops to values close to zero.
**5.  Summary and conclusions**
In this study, a simple model allowing the monitoring of the soil saturation level was
regionalized over the entire Iberian Peninsula, taking as a reference the soil moisture
simulated by a high-resolution land surface model. Two different regionalization methods
have been compared, either the direct estimation of soil water holding capacity from
european soil maps or by Random Forests, using covariates such as altitude,
temperature, precipitation, evapotranspiration and land cover. Results have shown that
the estimation by Random Forest is more robust notably to estimate low soil moisture
levels. Despite similar average performance between the two methods, the use of soil
maps to set the water holding capacity reveals less stable results in some cases, most
probably related to the uncertainties in the pedo-transfer functions used. While these
pedo-transfer functions are process-based predictive functions of certain soil properties,
Random Forest are not based on physical processes and are tailored to provide the best



estimates in a statistical sense. Therefore, they provide a valuable alternative in contexts
where high-resolution soil maps are not available since they rely on a set of covariates
that can be reliably estimated from global databases, such as satellite or reanalysis
products (Funk et al., 2015; Hersbach et al., 2020; Muñoz Sabater, 2020).
It should be noted that the results presented herein are highly dependent on the quality
of land surface simulations, in the absence of dense monitoring networks of in situ soil
moisture data, thus these results suffer from the same limitations as LSMs, notably, the
lack of human processes (irrigation). However, new remote sensing irrigation estimates
are being developed (Massari et al., 2021), as a consequence, once the RF model is
trained, irrigation estimations could be added to the precipitation forcing data in order to
include the human impacts on soil moisture estimations. The results show that this
approach allows us to cheaply extend the value of high resolution LSM simulations to
areas where no LSM is implemented (ie. north Africa), as long as the climate conditions
belong to the range of values used to train the model, mostly in terms of precipitation and
evapotranspiration ranges. Thus, the model train over the Iberian Peninsula could be
applied to other similar areas such as North Africa, Italy or Greece. As a perspective,
other simulations from countries where high resolution LSM simulations are available,
such as France or the USA, could be added to the database in order to expand the
coverage over different physiographic and climate contexts. Consequently, the benefits
of LSM simulations of soil moisture could be expanded to other areas, provided that
suitable forcing datasets are available. Furthermore, if public meteorological and
hydrological organizations were to create soil moisture observation networks, cleverly
designed to cover the most relevant climates of their countries, this approach could be
used to train the model using these observations and then regionalize the results to the
rest of the territory, thus, converting an *in-situ* observation dataset into a gridded dataset
with a much greater spatial coverage.

**Acknowledgements**
This work is a contribution to the HyMeX programme through the HUMID project
(CGL2017-85687-R, AEI/FEDER, UE).





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

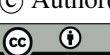


**TABLE**

Table 1: Contingency table of the comparison between forecasts and observations or
any two analyses. The symbols a–d are the different numbers of cases observed to
occur in each category.

|  | Observations | |
| --- | --- | --- |
| Forecast | 1 | 0 |
| 1 | a (hit) | b (false alarm) |
| 0 | c (miss) | d (correct rejection) |


**FIGURES**


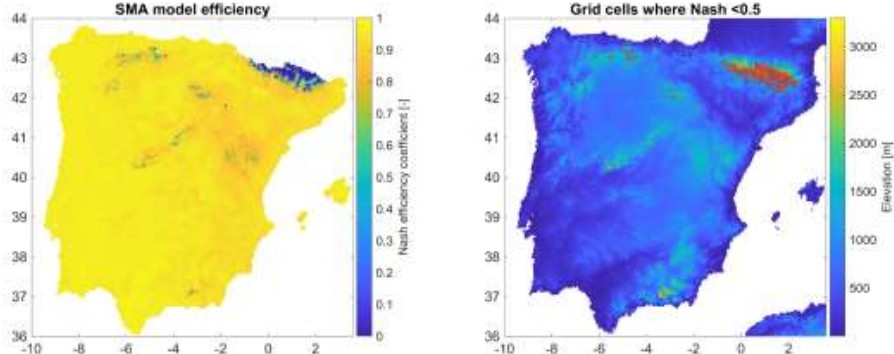


Figure 1: Efficiency of the SMA model to reproduce soil moisture from SURFEX





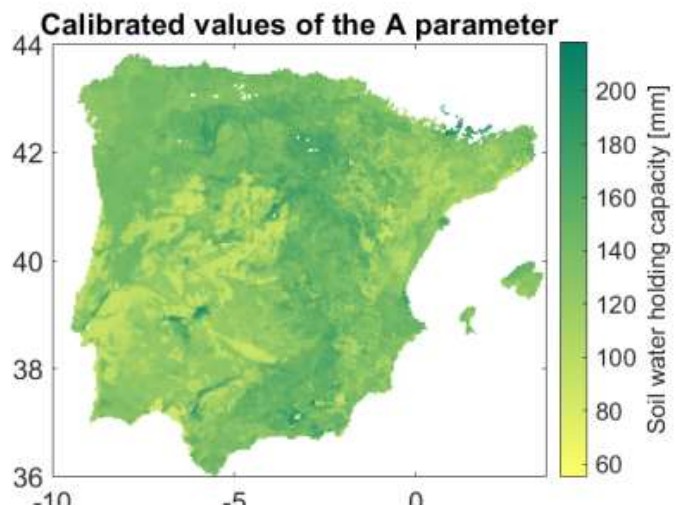

Figure 2: Map of the calibrated values of the A parameter of the SMA model

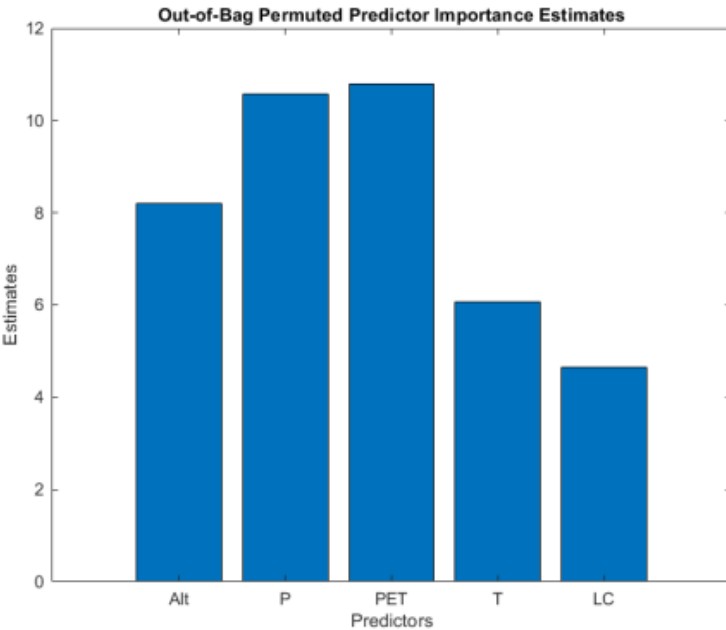

Figure 3: Relative importance of each predictor (Alt= altitude, P= precipitation, PET=
potential evapotranspiration, T=temperature, LC=land cover classes) in the Random
Forest method



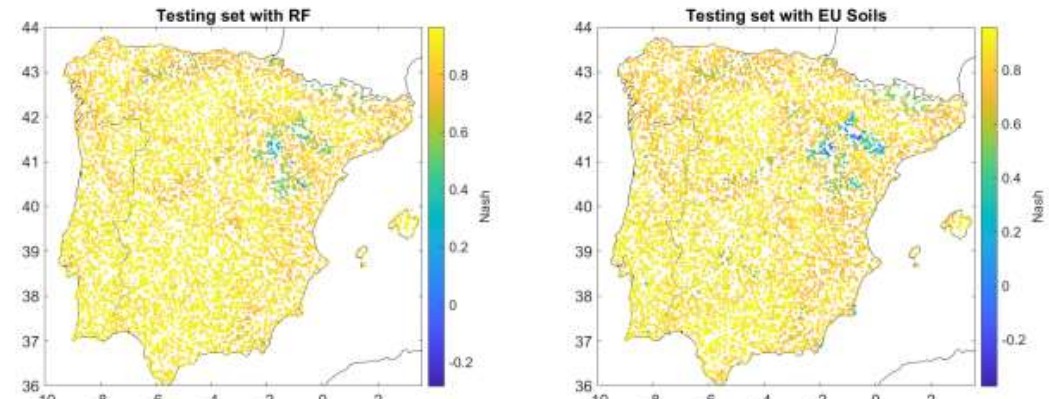

Figure 4: Nash efficiency coefficient obtained for the testing set, with the A parameter of
674                     the SMA model estimated by RF (left) or ESDB (right)


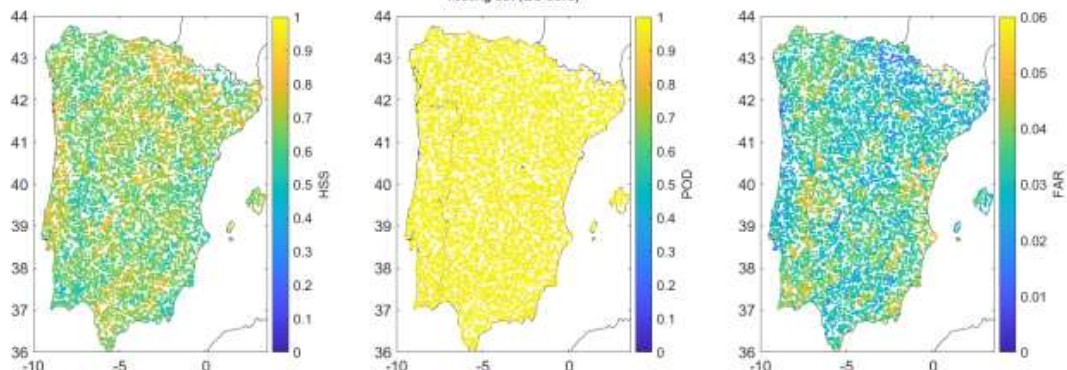

Figure 5: Validation results in terms of HSS, POD and FAR with A estimated with ESDB



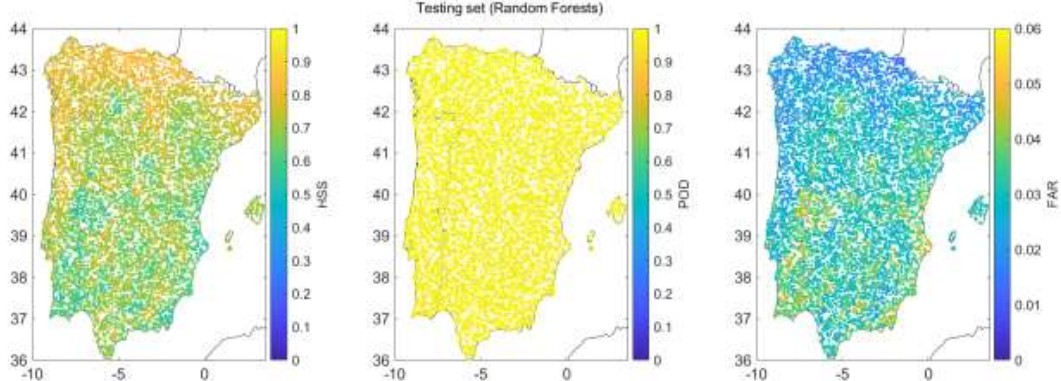

681           Figure 6: same as figure 5 but with A estimated with RF

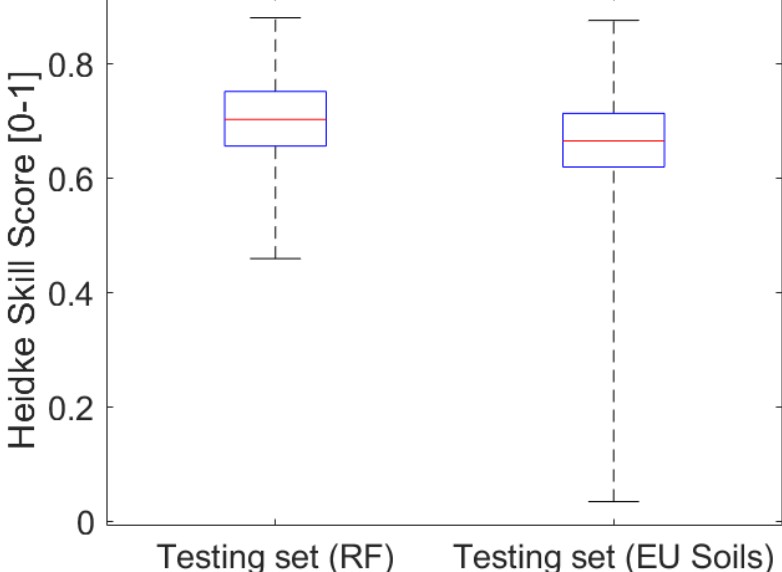

Figure 7: Boxplot of the HSS obtained with RF or EU soil maps. The limits of the box
represent the 25th and 75 percentiles, the line in the middle refers to the median, and
686          the limits of the whiskers extend to the minimum and maximum values.