# Peer review of "Estimating soil moisture conditions for drought monitoring with"

_Natural Hazards and Earth System Sciences, 2021_

## Author Response (AR1)

**REVIEWER 1**

**Main comments**

**This paper presents a interesting method to calculate a grid soil moisture index potentially on area not covered by a dense Land-surface models (LSM). This method is based on a simple 1-parameter soil moisture accounting model. This parameter is first optimized using as reference the temporal outputs of the SURFEX model, and then regionalized using Random Forest (RF) method and different grid static information (climatic variable and soil occupation).**

**The paper is potentially publishable in NHESS, after some minor corrections.In my view, the two main points that need to be improved are :**

**- the methods that are not always clearly described ;**

**- the results that are not sufficiently discussed.**

**The following detailed comments provid some change suggestions concerning those 2 points.**

We would like to thank you for this positive evaluation of our manuscript. We addressed all your comments, see the responses below and we modified the manuscript accordingly.

**Detailed comments**

**L1-2 : Maybe change the title in : « Estimating soil moisture conditions for drought monitoring with Random Forest and a simple soil moisture accounting scheme». See my very last rq**

Indeed, we had indeed some difficulties to find the most appropriate title and we agree with your suggestion. We modified the title.

**L28 : Is it « evapotranspiration » or « potential evapotranspiration » (see other similar rq)**

It is potential evapotranspiration, we modified everywhere it was required.

**L74 : « Stefan et al » not found in the bibliography, but « Vivien-Georgiana Stefan »**

We corrected the reference.

**L99-100 : Please remove « without calibration ». Regional methods are calibrated.**

we removed "(ie. to estimate from surrogate data without calibration)"

**L100-101 : The « water holding capacity » parameter and the « simple soil moisture model » should be briefly introduced before. Otherwhise, the reader has difficulties to understand the goal of the paper. Furthermore, I am not sure that the A parameter can be called « water holding capacity » (see rq later)**

We agree. This last paragraph of the introduction has been rephrased to:

The goal of the present study is to regionalize a simple soil moisture accounting (SMA) scheme that could be used to monitor soil moisture droughts. The SMA model considered in the present study requires a single parameter, the maximum soil water holding capacity. Two different approaches are compared to estimate this parameter regionally: the direct estimation with soil maps or with a machine learning technique, namely Random Forests.

**L123 : Is it « evapotranspiration » or « potential evapotranspiration » (see other similar rq). Furthermore, I am not sure that (P)ET is part of the SAFRAN variables. How did you obtain it?**

It is potential evapotranspiration (PET). We modified it everywhere it was needed in the manuscript. PET can be computed from SAFRAN variables using the Penman-Montheith equation. The Spanish SAFRAN dataset does not provide radiation data, for these variables we have used ERA5. Thus, PET is calculated using a combination of SAFRAN and ERA5 data using the Penman-Montheith equation.

**L132 : « Quintana-Segui et al 2020 » is not in the bibliography, but 2008 or 2019**

changed to Quintana-Seguí et al, 2019

**L140 : «Martinez-Fenandez et al 2015 » is not find in the bibliography, but « 2016 »**

changed to 2016

**L145 : I would introduce later the ESDB**

We don't understand this suggestion. This is the third paragraph of the section 2, "Study area and data", where we introduce all databases used in the present work: (i) the SAFRAN-spain database, (ii) the ECOCLIMAP2 database, and (iii) this ESDB database to retrieve soil properties. We believe this is the right place to introduce this database.

**L148-160 : I would move this part in « methods », when describing the first regionalisation method.**

We kept here the description of the data source (ESDB) but we moved into the method section the way this database was used.

**L159 : « Then, these estimates have been used to set the A parameter of the SMA model » should be remove. The SMA model and its A parameter should be introduced in « methods ». You should also maybe give the equation of your SMA (maybe in annex) otherwise it is hard to understand what exactly this « A parameter » stands for. And I'm not sure you can call it « soil water holding capacity » as you did L100. Its physical meaning is not so clear.**

We agree, this sentence is better suited in the method section, we moved it. We added the equation of the SMA model (from Perrin et al 2003, Tramblay et al., 2014, Javelle et al 2010) and we also expanded the description of the SMA model and included the equations. Since the

A parameter represents the maximum amount of soil water storage, it is assumed to be equivalent to soil water holding capacity.

**166 « PET » is not defined. Furthermore, you never talk about « potential evapotranspiration » before, but just « evapotranspiration » (see previous rq)**

We agree, as mentioned above, we replaced everywhere in the manuscript with 'potential evapotranspiration (PET)'.

**L166-174 : I would divide this part in two. One for describing ths SMA model, another to describe the « first regionalisation method ». In this second part I would place L148-160 (see previous rq). Furthermore I would clearly explain how do you make the link between A and TAWC ? Do you fit a regression model? Or simply assess A = TAWC ?**

We agree. We added a new section entitled "regionalisation of soil water holding capacity with soil databases". As mentioned above, we assess that A = TAWC.

**L174 : say clearly that you optimize A for each grid cell independently since « run » one SMA model per cell. This is not clear in your paper.**

Indeed, you are right that was not clear enough. We added that the calibration is done for each grid cell independently.

**L178-178 : not clear. Furthermore « SWI » is not introduced.**

We modified this sentence to: "The outputs of SURFEX soil moisture are first normalized with the maximum and minimum values. Then, the SMA model parameter A is calibrated using the normalized SURFEX soil moisture as a reference."

**L179 : give more details of this « direct estimation of A within TAWC (See previous rq). And do you say « direct » ? Because you state that A=TAWC ? Otherwise (regression) you should remove « direct » and « indirect » for RF. Both are regional methods with parameters to be calibrated, the later with much more parameters…**

As mentioned before, A = TAWC. This is now clearly explained in the method section.

**L181 : According to figure 3, RF does not use only « physiographic » variable, but also « climatic ».**

We agreed, and added "..and climatic" variables.

**L183 : change the title (without « water holding capacity »)**

We agreed and changed the title to be consistent with the other titles of this sub-section.

**L198 : I am not sure that the A parameter is called « soil water holding capacity »**

As mentioned before, A is assumed to be equivalent to soil water holding capacity. Since it is now better introduced in the methods, we removed this sentence here.

**L198 : « the properties of the 5x5 grid cells » : it is not clear. Which grid variable are used as input in your RF model?**

We changed to "using the physiographic and climatic properties, namely altitude, land cover, mean annual precipitation, temperature and PET, of each 5x5km grid cell"

**L208-212 : Same rq : which variable did you test together, how ? Figure 3 gives the final result, but we don't know how you reach it. If you only tested the variables from figure 3, just say it (it is not clear in the paragraph if you've tested more variables to finally selec only these ones). Furthermore, did you test your TAWC variable in your set of variable ?**

We tested the variables used in the random forest algorithm: altitude, land cover, mean annual precipitation, temperature and PET. We added this in the text.

We did not test TAWC since it is not one of the explanatory variables used in the random forest algorithm. The main goal of this procedure is to assess the relative influence of the different predictors, it makes little sense to include the predictand, TAWC, in the list of variables to be tested.

**L202 : Do you have some « hyperparameters » (ie structure of the model, optimization parameters...) in your random forest model ? If yes how are they chosen? (this is a naive question, I've never used random forest models, only neural networks, where you generaly define 3 periods : training, validation, test)**

There are indeed some hyperparameters that could be optimized such as maximum depth, maximum number of leaf nodes… Some preliminary experiments have been performed but due to the overall good efficiency of the method to estimate the A parameters in validation, the improvements due to hyperparameters tuning would be marginal.

It is true that for hyperparameter optimization it is required to have one training data set, to fit the parameters, a validation data set, to evaluate the quality of a model fit on the training data set while tuning hyperparameters and a test data set to provide a model evaluation on independent data. Since we did not implement hyperparameter optimization, we do not require a test data set.

**L253 : Do you mean Fig 2 instead of Fig 3 ?**

Yes, we changed to figure 2

**L263 : Please define clearly « each predictor ». Which ones ? Annual averaging ?**

altitude, land cover, mean annual precipitation, temperature and PET

**L263 : Does « cross validation » means « out of bag » used before in « methods » ? If yes please use only one term and cite the paragraph in « methods » explaining what have been done.**

Yes, we removed "cross validation" that is not appropriate here.

**L271-274: See previous rq on the tested input variables. In particular, was the TAWC also included in the set ? Any comment about the fact that climatic variables seems more important than «physiographic » (ie and use). Since P(t) and ETP(t) are the temporal inputs of the SMA model, one could think that A only depends on physiography.**

No, only the predictors are included to assess their relative influence on the estimation of A (considered equal to TAWC).

We added some comments about climate variables being more important than altitude or land cover.

**L276-278 : Don't repeat what is explained in the « methods », but just mention the paragraph. Furthermore, it is not clear if nash from Fig 4 are calculated using A regionalised with both methods versus A optimized with SURFEX, or using output from SMA (with A regionalised from both methods) versus rescaled SWI of SURFEX. I guess the first option is what is done.**

We changed the text to not repeat the methodology here.

The computation is done as option two, meaning we estimated A with Random forests or soil map, ran the SMA model, and compared the SMA simulation to SMI from SURFEX. That way, we evaluate the approach in pseudo-real conditions since we consider here the SURFEX soil moisture the variable that needs to be estimated and the closest to "real" soil moisture conditions.

**L288 : Put « Detection of dry soil moisture conditions» as in the title of 3.3**

We changed the title.

**L290-302 : The analysis could be more detailed. What is improved by the 2d method ? POD ? FAR ? Or both ? And where ? It could be maybe usefull to plot the (relative) differences (or ratio) between both methods to support the analysis. Furthermore, Figure 5 should be rescaled (setting asp=1 if you are using R)**

This is indeed a good suggestion to include a map of the difference between the two regionalization methods, we added a new figure 5, showing together the HSS, POD and FAR using either Random forests or EU soil maps, together with the difference between the two. The new maps show clearly the regions with the greatest improvements, being the north and northwestern parts of Spain, being the most humid, while little improvements are achieved in southern spain. We expanded the text about these new results:

"As shown in Figure 5, the results with Random forests mostly follow the climate conditions with improved estimations in the wettest regions of North and Northwestern part of Spain. For the estimation with EU soil maps, the results seem related to soil depth and to a lesser extent, land cover. Indeed, higher scores are found in regions with shallow soils, such as those of plutonic (Galician region, western parts of the Extremaduran mountainous ranges, Douoro basin) or metamorphic origins (western Cantabric range, north Iberian range, eastern-central regions and Sierra Morena in Andalucia) and also sedimentary regions with shallow limestones (eastern Cantabric mountains, Basque region, Southern Iberian range). On the opposite, lower scores are found in regions with the deepest soils (Guadalquivir floodplains, Mid- Tagus River,

upper Duero, piedmonts of Cantabric in Leon and Palencia, most of Middle Navarra). With the exception of regions such as Bizcaya or coastal Portugal, with a dense forest cover (mostly Pinus radiata or pinaster) where soil depth is probably overestimated."

**L306 : « soil saturation level » is used for the first time. Do you mean « soil moisture conditions » previously used ?**

Yes, we replaced 'level' by 'conditions'

**L323-344 : I agree with you on the usefullness of your method. But it is not only due to random forest, but also to the simple conceptual SMA model which with only one parameter per pixel, is able to « mimic » a physically based LSM. I think this main message should be reflected in the title (see first rq)**

We agreed and changed the title.

**REVIEWER 2**

**I have read with interest the manuscript, as well as the first reviewer's comments. I do not have much to add to the first reviewer's comments. I have a single minor comment related to the regionalization modelling procedure of the paper.**

Thank you for this positive evaluation of our work.

**In my understanding, the $A$ parameter is estimated by calibrating the SMA model against SURFEX soil moisture. Then RF are fitted to predict the $A$ parameter, and then the predicted $A$ parameter is set as input to the SMA model in a $70 - 30\%$ splitting scheme.**

This is correct.

**Thus for RF to work, they need a calibrated SMA model and SURFEX soil moisture (or similar products).**

Yes, the SURFEX soil moisture is used to be a benchmark to set up the Random forest estimation.

**Since SURFEX (or similar products) is a gridded product, the value of the modelling procedure is not clear to me. In particular, SURFEX or similar products needed to estimate the $A$ parameter, may cover densely the area, therefore, there is not clear to me why regionalization is needed.**

The main interest is not to apply our modeling approach where simulations from SURFEX or other similar land surface models are available, but to use it where such simulations do not exist. Of course, as for all modeling experiments, there is a need to assess first the relevance of the approach. This is why we produced a split-sample validation (on 30% of the data points) to estimate the reliability of soil moisture estimation with our approach in the absence of soil moisture simulated by SURFEX. This validation suggests that the approach is robust and provides similar soil moisture simulations as with SURFEX.

Once the A parameters are estimated with Random Forest, they can be then estimated outside of Spain using physiographic and climatic attributes only. As explained in the conclusions, other land-surface simulations could be used to expand the coverage on different physiographic and climatic contexts, and soil moisture measurements could serve to validate the approach outside of Spain. Also, if in the future there are more in-situ observations of soil moisture calculated with long enough series, the method could be used with in-situ data, instead of SURFEX. Furthermore, good quality satellite-based root zone soil moisture products may be produced in the future. With this method we could use these satellite data, instead of SURFEX, and then use the regionalized SMA model for time periods much longer than the satellite products.